# Global Contrastive Learning for Long-Tailed Classification

**Thong Bach**
thongbtqm@gmail.com

**Anh Tong**
**Korea Advanced Institute of Science & Technology**
anhtong@kaist.ac.kr

**Truong Son Hy**
**Indiana State University**
TruongSon.Hy@indstate.edu

**Vu Nguyen**
**Amazon**
vu@ieee.org

**Thanh Nguyen-Tang**
**Johns Hopkins University**
nguyent@cs.jhu.edu

**Reviewed on OpenReview:** https://openreview.net/forum?id=xWrtiJwJj5

## Abstract

We consider the long-tailed classification problem in which a few classes in the training data dominate the majority of the other classes. For concreteness, we focus on the visual domain in this paper. Most current methods employ contrastive learning to learn a representation for long-tailed data. In this paper, first, we investigate $k$-positive sampling, a popular baseline method widely used to build contrastive learning models for imbalanced data. Previous works show that $k$-positive learning, which only chooses $k$ positive samples (instead of all positive images) for each query image, suffers from inferior performance in long-tailed data. In this work, we further point out that k-positive learning limits the learning capability of both head and tail classes. Based on this perspective, we propose a novel contrastive learning framework that improves the limitation in k-positive learning by enlarging its positive selection space, so it can help the model learn more semantic discrimination features. Second, we analyze how the temperature (the hyperparameter used for tuning a concentration of samples on feature space) affects the gradients of each class in long-tailed learning, and propose a new method that can mitigate inadequate gradients between classes, which can help model learning easier. We name this framework as CoGloAT. Finally, we go on to introduce a new prototype learning framework namely ProCo based on coreset selection, which creates a global prototype for each cluster while keeping the computation cost within a reasonable time and show that combining CoGloAT with ProCo can further enhance the model learning ability on long-tailed data. Our code is available at **CoGloAT_ProCo**.

## 1 Introduction

Real-world data usually follows long-tail distribution, where only a few classes dominate the dataset (namely, head classes). In contrast, most other classes have a small number of samples (namely, tail classes). This long-tailed data poses a major inferential challenge to traditional deep learning models whose training is biased by the head classes and whose performance quickly deteriorates when the data is imbalanced (Wang

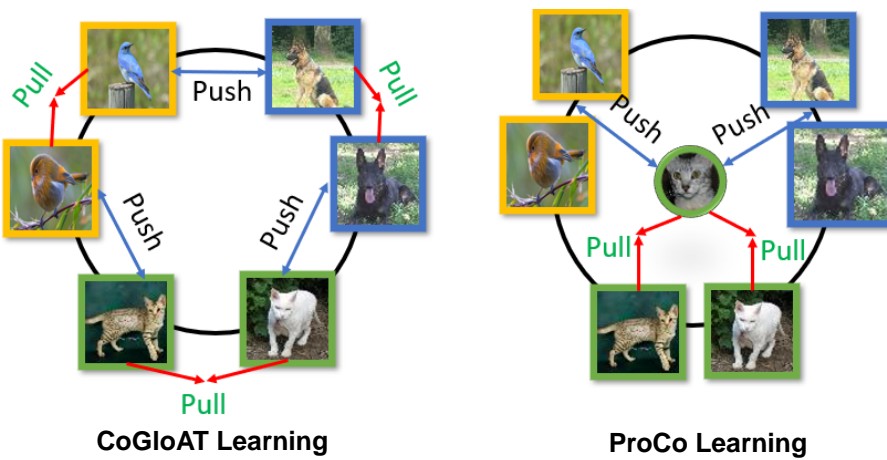

Figure 1: Overview of how CoGloAT and ProCo learn during training. CoGloAT learns to pull closer images of the same class while pushing apart images of different classes **across-batches**. Meanwhile, ProCo builds a prototype (center) for each class, by which points of the same class are pulled toward the class prototype whereas points of the other classes are pushed away.

et al., 2020; Cao et al., 2019; Zhang et al., 2021c). Various approaches have been proposed to address such issues, with typical methods including class-balanced re-sampling (Shen et al., 2016; More, 2016; Zhang et al., 2021c), class-level re-weighting (Alshammari et al., 2022; Lin et al., 2017; Zhang et al., 2021c) and ensemble learning (Zhou et al., 2020; Wang et al., 2020; Zhang et al., 2021b). However, the prior approaches rely on the classical cross-entropy losses which are sensitive to imbalanced data (Wang et al., 2020; Zhang et al., 2021c).

Many studies have shown that contrastive learning is more robust to noisy or unbalanced data (Khosla et al., 2020; Kang et al., 2020), therefore this algorithm is also widely applied to solve imbalanced data and achieves impressive results (Chen & He, 2021; Li et al., 2022b; Zhu et al., 2022; Liu et al., 2021; Yang & Xu, 2020). Its success is based on the contrastive loss function and a large number of negative samples (He et al., 2020), which will help the model learn more robust and semantic discrimination feature (Liu et al., 2021; Yang & Xu, 2020) thereby helping the model generalize well on the training data, and subsequently easily transfer to the test data or other data domains (Chen et al., 2020a; Liu et al., 2021). Moreover, unsupervised contrastive learning creates more balance feature space, even when the data is highly skewed (Kang et al., 2020), which is the main factor that results in the failure of the previous methods. However, the model trained in an unsupervised manner often does not perform well since it does not use label information; thus it fails to learn rich semantic discrimination features due to lack of label information. Different from unsupervised contrastive learning, supervised contrastive learning can learn more semantic feature space (Kang et al., 2020). Nonetheless, using the label during training makes features more imbalanced as compared with their unsupervised counterpart. To overcome this problem in supervised contrastive learning, instead of using all positive samples in the batch for training, Kang et al. (2020) proposes $k$-positive selection, a supervised contrastive learning method in which we limit the number of positive samples to $k$, therefore, it can reduce the imbalance effect. However, when the data is extremely long-tailed, using $k$-positive alone is not enough to tackle the imbalance problem (Li et al., 2022b). Besides contrastive learning, prototype learning (Li et al., 2022b; Cui et al., 2021; Zhu et al., 2022) is also widely used to tackle long-tailed problems. This method constructs a prototype (center) for each class and pulls samples of the same class closer to the center while pushing samples of other classes away. But it remains elusive as to how to build a good prototype and efficient prototype loss function.

Motivated by the limitations of the prior approaches, in this paper, we propose a training method namely **Co**ntrastive learning with $k$-**Glo**bal positive selection and **A**daptive **T**emperature (CoGloAT), which constructs a more balanced feature space with better semantic discrimination features. In CoGloAT, we first use label information to select $k$-positive samples across batches for the query. This so-called $k$-global positive

selection is in stark contrast with k-positive selection (Kang et al., 2020) which selects positive samples only within a batch. Expanding the selection space across batches helps the query connect to more positive samples; therefore the model can learn richer semantic discrimination features. Besides, we analyze the gradient of our model when training with imbalanced data, pointing out that the temperature value has an important role during the training. Based on the analysis, we propose a new method namely *Adaptive temperature*. In this method, we view temperature as a re-balanced parameter, which can help us reduce the imbalance effect between head and tail classes during the training stage by balancing their gradients. Moreover, we propose a prototype learning approach namely ProCo, which uses data distillation to efficiently construct prototypes. We empirically show that jointly training ProCo with CoGloAT can further improve the model performance.

In summary, our main contributions are a series of simple yet effective improvements that take a "global" perspective into contrastive learning to address the long-tailed (visual) classification (thus the name global contrastive learning):

- Contrastive learning with *k*-global selection and adaptive temperature (CoGloAT): We introduce a new training strategy that can adapt to any contrastive framework to improve its performance in long-tailed learning namely *k*-global positive selection. After that, we analyze the model gradient and propose re-balanced techniques using *adaptive* temperature to further tackle the imbalance problem. We empirically evaluate with different settings and show that our model outperforms the state-of-the-art model with a large gap.

- **Pro**totype learning via **co**reset selection (ProCo): We propose a new prototype learning method named ProCo. Compared to standard prototype methods, ProCo efficiently computes prototypes from a distilled coreset. Based on this coreset, calculated prototypes that are used in ProCo can represent global features for the original dataset without using all the training data.

## 2 Related work

Long-tailed recognition is one of the challenging problems in computer vision. There are many studies focusing on this problem with different solutions (Zhang et al., 2021c; Zhu et al., 2022; Shen et al., 2016; More, 2016; Alshammari et al., 2022; Lin et al., 2017; Zhu et al., 2022; Kang et al., 2020; Li et al., 2022b). The general idea is to try to re-balance the effect between head and tail classes during training from data perspective or model perspective.

**Data perspective** The most obvious way to solve long-tailed problems is to create a balanced dataset, where the samples of head and tail classes need to change to become more balanced. There are two popular approaches including over-sampling (Shen et al., 2016; Sarafianos et al., 2018), and under-sampling (More, 2016; Drummond et al., 2003; Buda et al., 2018). While the former simply creates more data for tail classes by duplicating samples, average samples, or learning its distribution then generating more data. The latter needs to reduce the samples of head classes by random removal procedures. However, the over-sampling approaches easily lead to overfitting (Bunkhumpornpat et al., 2012; More, 2016); under-sampling loss data information (More, 2016). To overcome the limitation of the two approaches, Park et al. (2022) introduces a new over-sampling method that creates larger tail samples by utilizing rich context features in head classes. Besides, we can use other data augmentation techniques (Li et al., 2021; Zhang et al., 2021d) which can give us a larger size for tail class, or data distillation techniques (Wang et al., 2018; Killamsetty et al., 2021; Ghadikolaei et al., 2019; Zhang et al., 2021a) to extract a smaller set for head classes while ensuring it can represent the distribution of the whole data.

**Model perspective** There are many techniques utilizing models to tackle long-tailed learning including re-weighting (Alshammari et al., 2022; Lin et al., 2017; Khan et al., 2017), ensemble learning (Zhou et al., 2020; Wang et al., 2020; Zhang et al., 2021b), or two stages training (Kang et al., 2019; Zhou et al., 2020). In re-weighting methods, we set weights for each class, usually reducing head class dominating by setting it small weight or enlarging the weight of tail classes to emphasize its contribution. The simplest way is to set weights

inverse to a number of samples in class. Besides, normalizing the weight of network (Alshammari et al., 2022) also shows an impressive result, or extends focal loss to Equalized Focal Loss (Li et al., 2022a) for efficient one-stage object detection. Along this direction, decoupling techniques use two stages of training, where we need to learn a good representation in the first stage and fine-tune it in the second stage. This training strategy shows that without carefully designing a model, we can also achieve good performance (Kang et al., 2019). Another typical method usually used in long-tailed learning is ensemble (Zhou et al., 2020; Wang et al., 2020; Zhang et al., 2021b), where we can joint-training a multiple branches network, each branch will contribute and learn specific information during training, then we combine the results of these branches together to predict the final result for long-tailed data.

**Contrastive learning** Contrastive learning has got attention recently because of its success in training for large unlabeled data (Chen et al., 2020a; He et al., 2020; Chen et al., 2020c; Doersch et al., 2015; Grill et al., 2020; Caron et al., 2018; Chen et al., 2020b; 2021; Tian et al., 2020), and labeled data (Khosla et al., 2020). In the decoupling method that we have mentioned above, the key problem is how can we train a strong backbone model at the first stage then use it as a pre-trained and fine-tune the whole or top layer of the model in the second stage. This is similar to the behavior of training contrastive model (He et al., 2020; Chen et al., 2020c;a; Chen & He, 2021), where we use a large amount of unlabeled data to learn invariant features from input data, then use those features to initiate for downstream tasks. Therefore, contrastive learning becomes a popular method in the first stage of training of decoupling. Recently, there have been an increasing number of studies trying to understand the behavior of contrastive learning on long-tailed data. For example, Kang et al. (2020) explore that self-supervised learning creates balance but lacks semantic feature space, and they propose $k$-positive learning which simply limits the number of positive samples in batch not greater than $k$ to ensure the balance between head and tail class. Liu et al. (2021) hypothesize that contrastive learning works better than supervised learning because it can transfer useful features from head classes to tail classes. Then they analyze the gradient and propose a method following theoretically-principled label-distribution-aware margin loss (LDAM)(Cao et al., 2019) to re-balance feature space. Zhu et al. (2022) show that supervised contrastive learning is not the optimal solution for long-tailed data and introduce class averaging and class complement in the loss function to make SupCon possible when training with long-tailed data. Zhou et al. (2022); Jiang et al. (2021) utilize the memorization effect of deep neural networks to recognize tail sample and enhance it in both data and model perspective.

**Prototypical contrastive learning** Training on self-supervised learning with long-tailed data generates more balanced feature space than the supervised counterpart. However, when the dataset is extremely skewed, it is still dominated by head class. Several works use prototype learning to re-balance feature space(Cui et al., 2021; Li et al., 2022b; Zhu et al., 2022) to overcome this issue. The key idea is to generate a set of prototypes (centers) for each class, then pull all samples of the same class closer to its prototypes and push samples from other classes far away. However, how to efficiently construct such prototypes is an active research area. Previous works (Li et al., 2020; 2022b) construct these prototypes by utilizing the large queue size in MoCo to save computation times. Nonetheless, it can not adapt to another contrastive framework, and when the dataset size is too large, data in the queue size can not represent the whole dataset distribution. In this paper, we propose another approach that can generate prototypes to cover the training data distribution without using all samples within an acceptable time.

## 3 Method

The training samples $S = \{(x_j, y_j)\}_{j=1}^N$ are i.i.d. drawn from a distribution over instance space $\mathcal{X}$ and label space $\mathcal{Y} = \{1, \ldots, C\}$ where $C$ is the number of classes. Our goal is to learn a function $f$ that maps the input data space $\mathcal{X}$ to the label space $\mathcal{Y}$ which minimizes the misclassification error $err(f) = \mathbb{E}_{(x,y)\sim\mathcal{D}}[\mathbb{1}\{f(x) \neq y\}]$. Inspired by previous works (Kang et al., 2020; Li et al., 2022b; Khosla et al., 2020), we propose a general framework that can apply to existing contrastive learning models e.g., MoCo (He et al., 2020), SimCLR (Chen et al., 2020a) to address the imbalance issue in long-tailed learning. Our model can create rich semantic features while reducing the dominating of head class during training. This framework includes two modules namely CoGloAT and ProCo. In the first section, we introduce Contrastive learning with k-global selection and adaptive temperature (CoGloAT), a method that can help contrastive learning models learn more easily

in the long-tailed setting. Then, we further improve Global contrastive learning by prototypical learning, where we propose another efficient method for prototype learning via coreset selection (ProCo).

### 3.1  Contrastive Learning with $k$-Global Positive Selection and Adaptive Temperature

### 3.2  $K$-global positive selection

SupCon (Khosla et al., 2020) and its improvement version for long-tailed learning: k-positive learning (Kang et al., 2020; Li et al., 2022b) select a positive sample from the same batch for each query sample. However, this selection strategy can limit the learning ability of the model, especially in tail classes, where the class size is small. More specifically, for an arbitrary class $i$, the probability that $k$ samples of a class $i$ appear in the same batch will be proportional to the number of samples in $i$. Hence with the tail class where the class size is usually small, it is more difficult to have enough $k$ positive samples in the same batch, and this makes the learning on these classes harder. To overcome this problem, we propose $k$-global positive selection. Our method expands the positive selection space for each query sample from within the batch to across batches. To implement the proposed $k$-global selection algorithm, we have redefined the DataLoader. Specifically, our DataLoader will include an additional variable containing information about the index of samples with the same labels in the training data. During training, for each sample $x$ in batch $B$, we will randomly select $k$ positive samples with the same label in training data (global selection) instead of choosing $k$ positive samples with the same label in batch (batch selection). This simple modification helps the query sample connect with all other positive samples of that class, therefore enlarging its selection space, and helping the model learn easier. Our formula for across-batches positive selection is defined as:

$$L_{\text{contrastive}} = -\frac{1}{N(k+1)} \sum_{i=1}^{N} \sum_{v_j^+ \in \{v_i^+\} \cup P_{i,k}^+} \ell(i,j), \tag{1}$$

where

$$\ell(i,j) = \log \frac{\exp\left(v_i \cdot v_j^+ / \tau\right)}{\exp\left(v_i \cdot v_j^+ / \tau\right) + \sum_{v_j \in V_i} \exp\left(v_i \cdot v_j / \tau\right)}.$$

Here $v_i^+$ is the augmented view of query $v_i$, $P_{i,k}^+$ is the k positive samples selected by k-global positive, and $V_i$ are examples in the current batch excluding $v_i$.

This selection strategy is simple yet extremely helpful for tail classes because, in long-tailed data, the total number of tail classes is usually small. Therefore if we can gather all the samples in tail classes together, each sample can utilize information from all other samples to learn, then our approach can efficiently learn the feature space of tail classes.

### 3.3  Adaptive Temperature

In this section, we discuss the gradient of the contrastive learning model, which has played an important role in classical machine learning to mitigate the imbalance effects (Tang et al., 2020; Ren et al., 2020). However, this result has not been explored in the contrastive setting. Therefore, based on the previous work of Wang & Liu (2021), we investigate its behavior in the context of the contrastive model and introduce a technique called *adaptive temperature* based on temperature value, which can effectively reduce the magnitude difference of gradients between head and tail classes.

**Gradient analysis**  Normalizing the gradient between head and tail classes has received much attention in cross-entropy learning (Tan et al., 2021). However, in contrastive learning, this problem has not been exploited much. As introduced in Khosla et al. (2020) with input $x_i$ and embedding function $f()$, the

gradient of embedding vector $z_i = f(x_i)$ is defined:

$$\frac{\partial L_i^{sup}}{\partial z_i} = \frac{1}{\tau} \left( \sum_{p \in P(i)} z_p \left( P_{ip} - \frac{1}{|P(i)|} \right) + \sum_{n \in N(i)} z_n P_{in} \right) \tag{2}$$

Here, $L_i^{sup}$ is the $L_{out}^{sup}$ in (Khosla et al., 2020), which is defined as:

$$L_{out}^{sup} = -\frac{1}{|P(i)|} \sum_{p \in P(i)} \log \frac{\exp z_i \cdot z_p / \tau}{\sum_{a \in A(i)} \exp z_i \cdot z_a / \tau},$$

where $\tau$ is a temperature parameter, $P(i)$ is a set of all positive samples of $z_i$ (set of all samples in a batch that have the same label with $z_i$), $N(i)$ is a set of all negative samples (set of samples have different class label with $z_i$), and $P_{ip} = \exp(z_i \cdot z_p / \tau) / \sum_{a \in A(i)} \exp(z_i \cdot z_a / \tau)$ where $A(i)$ is a set of all samples in the batch excluding $z_i$ (which means: $A(i) = P(i) \cup z_i \cup N(i)$).

In each batch, the gradient of a sample $i$ will be equal to the gradient of a positive sample ($P(i)$) plus the gradient of negative samples ($N(i)$) and then divided by temperature $\tau$. For easier analysis, set:

$$\frac{\partial L_i^{sup}}{\partial z_i^+} = \sum_{p \in P(i)} z_p \left( P_{ip} - \frac{1}{|P(i)|} \right) \tag{3}$$

$$\frac{\partial L_i^{sup}}{\partial z_i^-} = \sum_{n \in N(i)} z_n P_{in} \tag{4}$$

Now we can rewrite the Eq. 2 as the combination of positive gradient and negative gradient as:

$$\frac{\partial L_i^{sup}}{\partial z_i} = \frac{1}{\tau} \left( \frac{\partial L_i^{sup}}{\partial z_i^+} + \frac{\partial L_i^{sup}}{\partial z_i^-} \right) \tag{5}$$

Applying the above formulas to GloCo, we will have $|P(i)| = k$, which means that the number of positive samples for each query $x_i$ is fixed. The gradient of positive samples in Eq. 3 will be recalculated as follows:

$$\frac{\partial L_i^{sup}}{\partial z_i^+} = \sum_{p \in P(i,k)} z_p \left( \frac{\exp(z_i \cdot z_p / \tau)}{\sum_{a \in A(i)} \exp(z_i \cdot z_a / \tau)} - \frac{1}{k} \right)$$

where $P_{i,k}$ is the set of $k$ positive samples across batches of query $x_i$. Thus, choosing only $k$ positive samples during training will make the positive gradient of samples between different classes more balanced. This is another explanation for why KCL (Kang et al., 2020) works well on imbalanced data. However, as mentioned in Eq. 5, the gradient of embedding $z_i$ is a combination of positive and negative gradients. Choosing k positive samples helps to balance the positive gradient but does not balance the negative gradient. Expanding Eq. 4 for the negative gradient we have:

$$\frac{\partial L_i^{sup}}{\partial z_i^-} = \sum_{n \in N(i)} z_n \left( \frac{\exp(z_i \cdot z_n / \tau)}{\sum_{a \in A(i)} \exp(z_i \cdot z_a / \tau)} \right) \tag{6}$$

From this formula, when the negative sample is similar to the query (hard negative), its gradient becomes larger. Therefore the negative gradient will be contributed mainly from the hard negative samples. Besides, we can divide hard negative samples into two types: false negative (samples have the same class as $x_i$ but are considered negative) and true negative (similar samples of other classes). We have:

$$\frac{\partial L_i^{sup}}{\partial z_i^-} = \partial hard_{negative}^{false} + \partial hard_{negative}^{true} \tag{7}$$

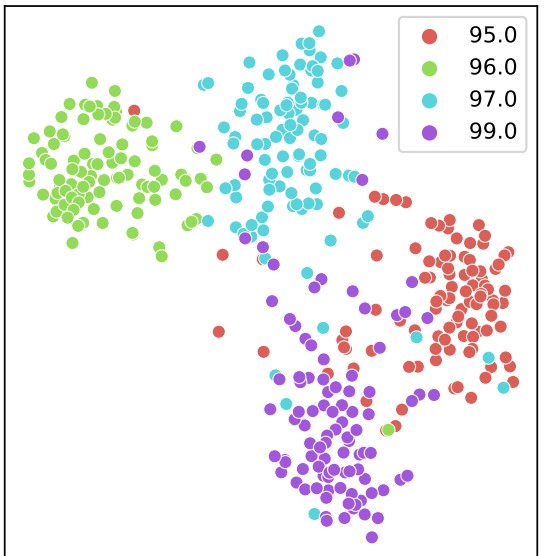 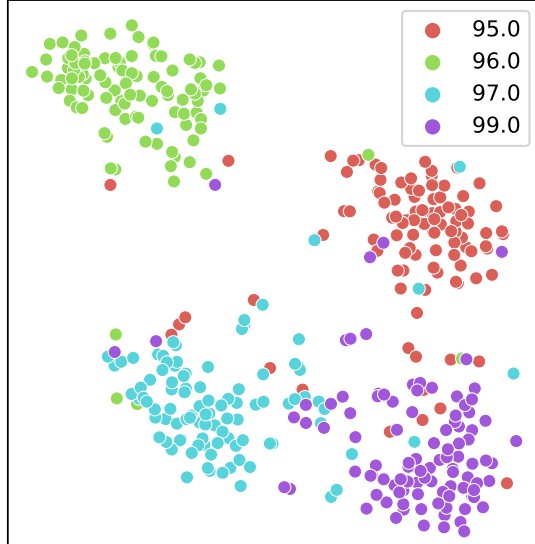

Figure 2: Feature space of four tail classes when using constant temperature (left) and adaptive temperature (right). Adaptive temperature helps the model learn more separable feature space.

In Eq. 7, while the true negative does not depend on the imbalance of the data, the false negative will be proportional to the imbalance factor, since classes with more samples (head class) will have more false negatives ( because we just choose $k$ positive samples for training, then the remained samples are considered as negative samples), resulting in the negative gradient of the head class being much larger than that of the tail classes, making training more difficult. Therefore, if we can remove the influence of false hard negatives, it will help the learning become more stable and balanced.

**Adaptive temperature formulation**  Work by  Wang & Isola (2020); Wang & Liu (2021) has shown that when the value of $\tau$ is small, the model will focus only on points near $x_i$ (hard negative), and as $\tau$ increases, the model will focus more on further points. Especially, when $\tau$ is large enough, the model considers all points (normal negative and hard negative) to contribute equally. Therefore in head classes, to reduce the influence of false-hard negative samples, we should increase the value of $\tau$ and vice versa, for tail classes, because the number of false-hard negatives is not large, the reduction of $\tau$ will not affect the negative gradient too much. Besides, $\tau$ can be viewed as a re-weighted parameter in Eq. 2, where tail classes with small $\tau$ will enlarge their gradient and vice versa. Based on these observations, instead of using temperature as a constant parameter, we can assign different temperature values to samples of different classes with the property that head classes will have large $\tau$ values and tail classes will have smaller $\tau$ values. To ensure the above property, we propose a formula where the temperature of each class will be proportional to the number of samples of that class:

$$\tau_i = \gamma + (1-\gamma) \cdot \frac{\text{frequent}(\text{class}_\text{i})}{\text{frequent}(\text{class}_\text{max})} \tag{8}$$

where $\tau_i$ is the temperature of class $i$, $\gamma$ is the minimum value of temperature, $frequent(class_i)$ is the number of samples in class $i$, and $frequent(class_{max})$ is the number of samples of largest class.

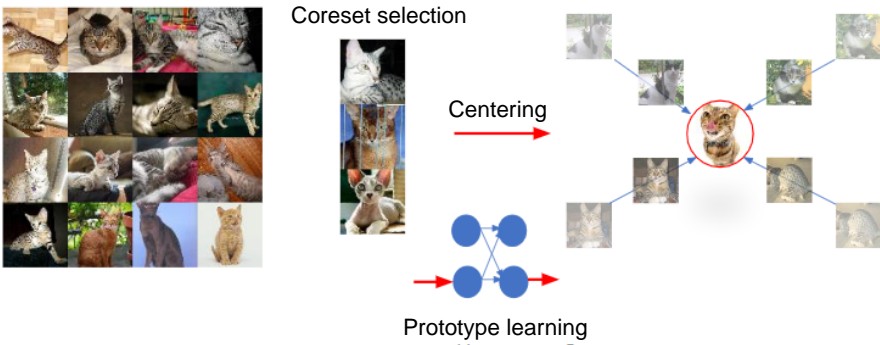

Figure 3: Prototype learning via coreset selection pulls images closer to their centers. This will encourage the samples in the same class to be closer together, thus increasing semantic discrimination feature space.

### 3.4 Prototype Learning via Coreset (ProCo)

Previous works (Li et al., 2020; Zhu et al., 2022) determine prototype as a mean representation of feature space for each class. During the training, these prototypes are viewed as positive samples in the contrastive loss. Therefore, prototype training is similar to clustering (Li et al., 2020), where it pulls samples in the same class closer to its prototype (the role of a prototype here is like a centroid of class) while pushing samples from another class far away. Recently, a few applications using prototype learning have been proposed to mitigate imbalanced data problems. TSC (Li et al., 2022b) re-balance feature space by pre-defining balanced targets, then moving samples in class close to their assigned balanced prototypes based on the matching algorithm. BCL (Zhu et al., 2022) analyzes the behavior of supervised contrastive learning, then proposes two new concepts named: class-averaging, and class-complement based on prototypes to mitigate the imbalance effect in supervised contrastive learning. In this paper, we propose another approach to prototype learning. Firstly, we tackle the trade-off between global prototype representation and computation cost (the total running time for calculating prototype), our method achieves global mean representation for each class while saving significant computation cost due to the estimation mean being conducted on the coreset (the set contains important samples of the training data, where its size is significantly smaller than the training set). This makes our prototypes globally represent the feature in data without using all training samples. Secondly, we propose a new prototype loss function and empirically show our loss function can work well on imbalanced data and improve the contrastive learning model significantly. The detailed effect of this module is shown in Table 5.2

**Coreset selection for calculating prototype** The main challenge in prototype learning is how can we efficiently calculate prototypes, both in computation time and representation ability. Previous works determine prototypes from the momentum queue feature in MoCo (Li et al., 2020; 2022b), which can yield consistent prototypes with low computation costs. However, this method can not adapt to another contrastive learning model that does not use a large queue size (Chen et al., 2020a). Besides, when the dataset size is extremely large (Krizhevsky et al., 2017), calculating the prototypes based on samples in the queue can yield local prototypes. To solve the above limitations, we distill the original dataset to extract its most important samples, then we will calculate prototypes according to this set. Our algorithm is as follows: we first use coreset generation algorithm (Killamsetty et al., 2021) as a data distillation method to extract the coreset. During the training, we update our prototypes based on the coreset at the beginning of each epoch. Calculating prototypes from the coreset encourages the centers to represent their class globally, while our method can easily adapt to any framework and dataset size.

**Prototype loss function** After having a coreset, we augment two views of it and calculate two prototype sets for every two views. Then we define a symmetric prototype loss function, based on contrastive learning loss. This loss function uses the calculated prototypes as the positive sample for a query having the same

class label. Assume each query $v_i$ augment two views $v_{i,1}^+, v_{i,2}^+$. We define our loss as the contrastive loss between each pair of the query sample and its prototypes:

$$-\frac{1}{4N} \sum_{i=1}^{N} \sum_{v_+ \in \{v_{i,1}^+, v_{i,2}^+\}} \sum_{q \in \{1,2\}} \log \frac{e^{\frac{v_+ \times c_{i,q}}{\tau}}}{e^{\frac{v_+ \times c_{i,q}}{\tau}} + \sum_{c \in C_q^-} e^{\frac{v_+ \times c}{\tau}}},$$

where $c_{i,q}$ is the prototype (center) of query $i$ in coreset $q$, $v_i$ is calculated from coreset $q$, and $C_q^-$ are the list prototypes from coreset $q$ excluding $c_{i,q}$. Our proposed prototype formulation is different from others (Li et al., 2022b; Zhu et al., 2022) in its symmetric loss between different views of query and coreset, which can help the model learning achieve more stability.

**Joint training CoGloAT and ProCo**  The final loss function will be the weighted combination between CoGloAT and Proco, which is defined as:

$$L_{final} = \alpha L_{CoGloAT} + (1 - \alpha) L_{ProCo}$$

where $\alpha$ is the hyperparameter that weights the CoGloAT and ProCo loss during training, which will be tuned during the training.

## 4   Analysis

To further understand the benefit of our model, we extensively evaluate several aspects of the contrastive learning model based on two important characteristics: *Alignment* and *Uniformity*. Besides, we discuss the limitations of the current formula for calculating Intra Alignment and Inter Uniformity, followed by our improved version that can be calculated more efficiently. Finally, we evaluate these characteristics in our models to further understand their behavior when learning on long-tailed data.

**Intra Alignment**  This metric is the average distance between all samples in the same class on feature space (Li et al., 2022b). It is similar to a measure of dispersion: **variance**, meaning when the alignment value is small, samples in the same class will lie close to each other, and vice versa. The alignment formula is defined as:

$$\text{Intra} = \frac{1}{C} \sum_{i=1}^{C} \frac{1}{\|F_i\|^2} \sum_{v_j^i, v_k^i \in F_i} \|v_j^i - v_k^i\|^2. \tag{9}$$

where $F_i$ is the feature space of samples in class $i$, $(v_j^i, v_k^i)$ are the two arbitrary feature vectors in feature space $F_i$. This formula is a good choice to estimate the spread out of samples in a class. However, it takes a lot of time to calculate. For example, with a class has $n$ samples, the time complexity of Intra Alignment is $\mathcal{O}(n^2)$, so when the number of samples in a class is large, it is computationally expensive to calculate Intra Alignment. To overcome this issue, we first need to construct centers for each class, then calculate the variance by:

$$\text{Variance} = \frac{1}{C} \sum_{i=1}^{C} variance_i \tag{10}$$

In this formula, $variance_i$ is the variance of class $i$ and can be calculated by:

$$variance_i = \frac{1}{F_i} \sum_{v_j^i \in F_i} \|v_j^i - c_i\|^2 \tag{11}$$

where $c_i$ is the center of class $i$. This reduces the computational cost from $\mathcal{O}(n^2)$ to $\mathcal{O}(n)$.

| Metric | | CIFAR-10-LT | | | CIFAR-100-LT | | |
|---|---|---|---|---|---|---|---|
| | | Imbalance factor | | | | | |
| | | 100 | 50 | 10 | 100 | 50 | 10 |
| Intra | KCL | 0.43 | 0.42 | 0.42 | 0.55 | 0.57 | 0.45 |
| | Ours | **0.41** | **0.41** | **0.39** | **0.44** | **0.43** | **0.41** |
| Variance | KCL | 0.65 | 0.62 | 0.61 | 0.84 | 0.85 | 0.63 |
| | Ours | **0.61** | **0.60** | **0.57** | **0.66** | **0.65** | **0.60** |
| Inter | KCL | 0.70 | 0.75 | 0.83 | 0.90 | 0.95 | 0.93 |
| | Ours | **0.73** | **0.79** | **0.85** | **0.98** | **1.00** | **0.98** |
| Improved Inter | KCL | 1.10 | 1.25 | 2.03 | 1.91 | 2.00 | 2.31 |
| | Ours | **1.20** | **1.29** | **2.20** | **2.23** | **2.36** | **2.41** |

Table 1: We measure two common characteristics of feature space, which have been introduced in section 4: **Intra Alignment** and **Inter Uniformity**. These measurements are conducted on the learned features using our model (**Ours**) and the baseline KCL (**KCL**) on two different datasets: CIFAR-10-LT and CIFAR-100-LT with different imbalance factors (the ratio between largest and smallest class in terms of frequent) range in $\{10, 50, 100\}$. From these results, we can highlight two insights: **(i)** Our proposed measurements: $Variance$ and $Inter_{improved}$ yield better results (in terms of computation cost for alignment and accuracy for uniformity) compared with the original formulas, **(ii)** Training with CoGloAT + ProCo creates a more robust feature space (it returns smaller Intra Alignment and larger Inter Uniformity values compared with KCL).

**Inter Uniformity** Different from Intra Alignment, Inter Uniformity is used to measure the distance between classes (Li et al., 2022b). It is the total distance between the centers of all classes. When Inter Uniformity is large, it means feature space is more separable, and vice versa. The original formula of Inter Uniformity is defined as:

$$\text{Inter} = \frac{1}{C(C-1)} \sum_{i=1}^{C} \sum_{j=1, j \neq i}^{C} \|c_i - c_j\|^2 \tag{12}$$

where $c_i$ represents as a mean feature vector of class $i$ based on its feature space $F_i$, and we can achieve it via: $c_i = \sum_j \frac{1}{|F_i|} v_i^j$, $j$ is the index of feature vector in $F_i$. This formula however has a caveat: it does not use the alignment information, and consequently does not always reflect the intended meaning of "inter uniformity". A concrete example is when we have two pairs of classes, that have the same center distance, but in the first pair each class has a high variance, and in the second pair each class has a low variance. The Inter Uniformity of the second pair should be larger because the boundary between classes is larger, but with the above formula, it will return the same result. Therefore, we modify the original uniformity formula by adding variance information to capture the above situation. The proposed formula is follows:

$$\text{Inter}_{\text{improve}} = \frac{2}{C(C-1)} \sum_{i=1}^{C} \sum_{j=1, j \neq i}^{C} \frac{\|c_i - c_j\|^2}{|\text{variance}_i + \text{variance}_j|}. \tag{13}$$

where $\text{variance}_i$ is defined in Eq. 11.

The above formula estimates the uniformity of data better when combined with alignment information. Table 1 shows the effectiveness of this formula, while the original uniformity formula indicates that **KCL** has a similar uniformity value with CoGloAT + ProCo in some settings, its improved version: $Inter_{improved}$ yields a more consistent result, which show by CoGloAT + ProCo gives a higher uniformity in all settings. The results from the improved uniformity formula are more reasonable because prototype learning helps the model learn more separable feature space, therefore increasing its uniformity.

## 5 Experiments

### 5.1 Dataset and implementation details

**CIFAR-10-LT and CIFAR-100-LT** For small-scale data, we conduct experiments on CIFAR-10-LT and CIFAR-100-LT .[1]The original CIFAR-10 and CIFAR-100 contain 50,000 32x32 images for the training and 10,000 32x32 images for the testing. The label of both datasets follows a uniform distribution, where each class has the same number of samples. There are 10 classes in CIFAR-10 and 100 classes in CIFAR-100, respectively. Based on this data, its imbalance versions: CIFAR-10-LT and CIFAR-100-LT are created following (Cao et al., 2019), where we create the imbalance training data while keeping the test data unchanged. The imbalance training data is defined through the imbalance ratio $p = max(n_i)/min(n_i)$ between the most frequent and least frequent size classes. Long-tailed imbalance follows the exponential decay in sample size across classes. In this paper, we run the experiment under $p \in \{10, 50, 100\}$

**ImageNet-LT** This is a long-tailed version of ImageNet dataset (Deng et al., 2009), where the long-tailed data is constructed by selecting the subset following the Pareto distribution with power value $\alpha = 6$. Basically, this dataset contains 115,800 of 224x224 images from 1000 different classes, where the largest class has 1280 images and the smallest contains 5 images. And similar to the CIFAR data, the test set of ImageNet-LT is the same as its original test version.

**Implementation details** Similar to TSC (Li et al., 2022b), we use Mocov2 (a common self-supervised learning algorithm (Chen et al., 2020c) here with the same configurations in TSC. For **CIFAR** data, we use batch size 256, initial learning rate 0.1, SGD optimizer with momentum 0.9, and we train the ResNet32 as the backbone models for 1,000 epochs. After pre-training on our framework, we fine-tuned the top-classifier layer of the pre-trained model with LDAM (Cao et al., 2019) loss (motivated by minimizing the margin-based generalization bound) and Reweight (Wang et al., 2017) (the weight of each class is proportionally inverse to its frequent) for CIFAR-10-LT, and Cross-Entropy loss combine Reweight for CIFAR-100-LT. Both of them use the same learning rate equal to 0.1 and a batch size equal to 256. We conduct the experiments on three different imbalance factors $\{10, 50, 100\}$, where a larger imbalance factor means the data is challenging to learn. The result in Table 5.2 shows that our model can outperform the current state-of-the-art model on all datasets across a range of imbalance factors by a large margin. For **ImageNet-LT**, its training strategy is similar to CIFAR data, we just change the backbone from ResNet-32 to ResNet-50, this model is trained on 200 epochs before fine-tuning the classifier layer with an additional 100 epochs before making the final prediction. The performance of our model with the ImageNet-LT dataset can be found in Table 5.2

### 5.2 Result

**CIFAR-10-LT and CIFAR-100-LT** Table 5.2 shows that our model outperforms the current state-of-the-art model on both datasets with different imbalance factors. In particular, on CIFAR100-100-LT, the model increased by approximately 3%, compared to the best model: TSC (Li et al., 2022b) (50.4% vs 47.4%). This improvement comes from the efficient extraction of positive samples from the k-global positive selection and unsupervised loss function, helping the model learn balanced and semantic features. Besides, CoGloAT alone already outperforms TSC (Li et al., 2022b) (k-positive + prototype learning). This suggests that CoGloAT is a stronger baseline in contrastive learning when solving long-tailed problems.

**ImageNet-LT** In table.5.2, we compare the performance of our model with other baselines on the ImageNet-LT dataset. Our model achieves a slight improvement of 0.3% over the baseline (KCL). This modest gain is likely due to the fact that we have not performed any hyperparameter optimization/selection for the ImageNet-LT dataset; instead, we use the same hyperparameters as on the CIFAR-100-LT dataset. [2] We expect that fine-tuning our model on ImageNet-LT will lead to a more significant improvement.

---

[1]For each setting, we evaluate it with 3 seeds and report its accuracy and standard deviation. The results of other methods are extracted from TSC (Li et al., 2022b) which do not have a standard deviation.

[2]Due to our resource constraints.

| Method | CIFAR-10-LT | | | CIFAR-100-LT | | |
|---|---|---|---|---|---|---|
| | Imbalance factor | | | | | |
| | 100 | 50 | 10 | 100 | 50 | 10 |
| CE | 70.4 | 74.8 | 86.4 | 38.3 | 43.9 | 55.7 |
| CB-CE (Cui et al., 2019) | 72.4 | 78.1 | 86.8 | 38.6 | 44.6 | 57.1 |
| Focal (Lin et al., 2017) | 70.4 | 76.7 | 86.7 | 38.4 | 44.3 | 55.8 |
| CE-DRW (Cao et al., 2019) | 75.1 | 78.9 | 86.4 | 40.5 | 44.7 | 56.2 |
| LDAM (Cao et al., 2019) | 73.4 | 76.8 | 87.0 | 39.6 | 45.0 | 56.9 |
| LDAM-DRW (Cao et al., 2019) | 77.0 | 80.9 | 88.2 | 42.0 | 46.2 | 58.7 |
| M2m-LDAM (Kim et al., 2020) | 79.1 | - | 87.5 | 43.5 | - | 57.6 |
| PCL (Cui et al., 2021) | - | - | - | 52.0 | 56.0 | 64.2 |
| BCL (Zhu et al., 2022) | 84.3 | 87.2 | 91.1 | 51.9 | 56.6 | 64.9 |
| KCL (Kang et al., 2020) | 77.6 | 81.7 | 88.0 | 42.8 | 46.3 | 57.6 |
| TSC (Li et al., 2022b) | 79.7 | 82.9 | 88.7 | 43.8 | 47.4 | 59.0 |
| **CoGloAT** | 81.4±0.1 | 84.5±0.4 | 88.9±0.1 | 45.4±0.1 | 49.5±0.1 | 60.7±0.1 |
| **CoGloAT + ProCo** | **81.6**±0.3 | **85.5**±0.1 | **89.4**±0.2 | **46.0**±0.2 | **50.4**±0.2 | **60.8**±0.1 |

Table 2: The top-1% acc run on CIFAR-10-LT and CIFAR-100-LT with Resnet-32 backbone. In this table, the accuracy of CoGloAT + ProCo outperforms the previous state-of-the-art model by a large margin in all experiments, especially on CIFAR-100-LT, which increases 3% (on imbalance factor 50). This result suggests that our model can be viewed as a new baseline for contrastive learning in long-tailed recognition.

| Method | Many | Medium | Few | All |
|---|---|---|---|---|
| OLTR (Liu et al., 2019) | 35.8 | 32.3 | 21.5 | 32.2 |
| LWS (Kang et al., 2019) | 57.1 | 45.2 | 29.3 | 47.7 |
| PCL (Cui et al., 2021) | - | - | - | 57.0 |
| BCL (Zhu et al., 2022) | - | - | - | 56.0 |
| KCL (Kang et al., 2020) | 61.8 | 49.4 | 30.9 | 51.5 |
| TSC (Li et al., 2022b) | 63.5 | 49.7 | 30.4 | 52.4 |
| **KCL†** | **59.5** | **50.3** | 38.5 | 49.5 |
| **Ours** | 59.0 | 49.5 | **40.9** | **49.8** |

Table 3: The top-1% acc run on ImageNet-LT. Here KCL† means the baseline model that we have reproduced.

| Model | CIFAR-10-LT | | | CIFAR-100-LT | | |
|---|---|---|---|---|---|---|
| | Imbalance factor | | | | | |
| | 100 | 50 | 10 | 100 | 50 | 10 |
| KCL | 12.0 | 14.0 | 21.0 | 14.5 | 15.5 | 22.5 |
| Global selection | 28.5 | 34.0 | 50.5 | 34.0 | 37.5 | 57.5 |
| Adaptive temperature | 12.5 | 14.5 | 22.0 | 15.5 | 16.5 | 22.5 |
| ProCo | 22.0 | 25.0 | 35.0 | 26.0 | 27.5 | 36.0 |

Table 4: The running time (in seconds) of different settings for one training epoch. Global selection increases the time proportionally to the number of k (we choose k=3 in our model) positives, while ProCo adds extra time to update the class centers after each epoch and compute the prototype loss.

| Loss type | CIFAR-100-LT | | |
|---|---|---|---|
| | Imbalance factor | | |
| | 100 | 50 | 10 |
| Adaptive | 43.7 $\pm$0.1 | 48.6 $\pm$0.2 | 57.9 $\pm$0.1 |
| Proco | 44.3 $\pm$0.1 | 47.8 $\pm$0.1 | 58.7 $\pm$0.2 |

Table 5: The accuracy of Adaptive temperature and ProCo alone on CIFAR-100-LT dataset. Each of these modules independently improves the accuracy of the model. Therefore, it is possible to combine various model architectures with one of these techniques: $k$-global selection, adaptive temperature, or ProCo to improve their performance.

### 5.3 Ablation study

**Running time of different modules**  Contrastive learning models can benefit from adding $k$-global selection or ProCo, but these modules also increase the running time. To better understand the efficiency of our model when being applied to different scenarios, we measure the running time of the baseline model when it is trained jointly with $k$-global positive selection, adaptive temperature, and ProCo, individually. The detailed result is shown in Table 4.

**How does each module contribute to the baseline?**  We evaluate the contribution of each module by combining it with the baseline model **KCL** (Kang et al., 2020): adaptive temperature, and prototype learning. Training with these modules individually improves learning accuracy. Table 5.3 shows the detailed results.

**How do different loss functions contribute to the model performance?**  In the $k$-positive selection, sampling positive samples in the queue of MoCo leads to a trade-off between equalizing positive samples across classes, and semantic discrimination of the learned features. Reducing the value of $k$ makes the model learn more balanced features but reduces the quality of the model, and vice versa. In CoGloAT, to avoid this trade-off, we construct two independent modules: while $k$-global positive selection has the role of learning semantic discrimination features, an unsupervised contrastive loss is responsible for balancing a learned space between classes. This learning strategy helps the model learn more efficiently. In Table 6, we show the behavior of CoGloAT when using three different loss functions: unsupervised, supervised, and supervised contrastive losses with $k$-positive selection. While the model learns with unsupervised loss (the default loss function in CoGloAT) has the best accuracy, combining it with $k$-positive loss yields an accuracy higher than the supervised setting. Besides, CoGloAT + $k$-positive loss outperforms the original $k$-positive version by a large gap (1.3% - 1.5%). This suggests CoGloAT to be a strong baseline for other methods in long-tailed learning.

| Loss type | CIFAR-100-LT | | |
| --- | --- | --- | --- |
| | Imbalance factor | | |
| | 100 | 50 | 10 |
| Supervised | 43.7 ±0.1 | 48.6 ±0.2 | 57.9 ±0.1 |
| K-positive(KCL) | 44.3 ±0.1 | 47.8 ±0.1 | 58.7 ±0.2 |
| Unsupervised | **45.4** ±0.1 | **50.4** ±0.2 | **60.8** ±0.1 |

Table 6: The accuracy of CoGloAT with different loss types. Unsupervised contrastive loss outperformance its supervised counterparts

## 6 Conclusion and Future work

**Conclusion** In this paper, we gave an overview of the current works on long-tailed data with contrastive learning, both its performance and limitations. After that, we introduced two new methods named CoGloAT and ProCo which can improve the training of contrastive learning models for long-tailed data in different ways. For each method, we created a stronger model built upon leveraging the advantages of the previous method while improving its limitations. Then we conducted a variety of experiments to highlight the contribution of each method. Finally, we reviewed the previous evaluation metric in contrastive learning, discussed its remaining problems, and proposed our improved version.

**Future work** Our model solely tackles data imbalance based on the general characteristics of contrastive learning. It is advantageous to build a robust baseline model like CoGloAT. However, combining contrastive learning with other methods to mitigate data imbalance such as class sampling, class re-weighting, logit adjustment, etc. has not been explored thoroughly in our work. Therefore, it is an interesting direction to include the above methods in CoGloAT to have both theoretical and experimental observations. Another avenue for future work is to develop a rigorous theoretical framework to explain the benefits of our contrastive learning framework for general long-tailed classification problems.

## Acknowledgments

We would like to thank Nhat Khang Ngo for his help in running some of our large-scale experiments.

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
