# OpenReview forum: "Global Contrastive Learning for Long-Tailed Classification"
_TMLR — Accepted by TMLR_

### Review · Reviewer_VF6P · 2023-06-05

**Summary Of Contributions:**

The paper aims to solve the long-tailed visual classification problem via contrastive learning. Specifically, the authors propose to sample $k$ positive samples for each query from the whole training set instead of a batch and adaptively balance the gradients by adjusting the temperature. Moreover, they propose a prototype learning method based on the coreset.

**Audience:**

Yes

**Broader Impact Concerns:**

NAN

**Claims And Evidence:**

No

**Requested Changes:**

Please refer to the Weaknesses part.

**Strengths And Weaknesses:**

Strengths:
1. The motivations are clearly described;
2. The performance improvement is overall significant.

Weaknesses:
1. Some core descriptions of the proposed techniques are missing, e.g., how to perform the $k$-global positive selection, and the definition of $L_i^{sup}$.
2. It seems that Prop. 1 is insufficient to explain the proposed $k$-global positive selection. What is the relationship between the so-called batch/global balance ratio and the long-tailed classification performance?
3. Although the proposed adaptive temperature (Eq. (13)) sounds reasonable, it is unclear how it balances the gradients.
4. Experiments are conducted on only small-scale datasets, CIFAR-10 and CIFAR-100. It will be more convencing if experiments on large-scale datasets like imagenet and Inaturalist are provided.

---

> ### Author Response · Authors · 2023-06-21
> **We thank the reviewer for your positive comment. We would like to address your concerns as below:**
>
> 1. We have updated the missing descriptions about how to implement $k$ global selection in section 3.1 and the definition of $L_i^{sup}$ in section 3.2, which are highlighted in blue color. In summary, we implement k-global selection via defining a new data loader, while $L_i^{sup}$ is actually the supervised contrastive learning loss.
> 2. Our \textbf{balance ratio} inspires by the imbalance factor in long-tailed data, where a higher imbalance factor means the data is highly skew to some classes, and reduces the long-tailed classification performance. In our setting, the \textbf{balance ratio} represents \textbf{the model's ability to mitigate the imbalance in training data} through how the algorithm selects $k$ positive samples (which we have defined as selection space). And similarities to the imbalance factor, a smaller balance ratio leads to better the model's accuracy. Consequently, our proposed method: $k$-global positive selection proves that it can reduce the balance ratio (which is defined as \textbf{global balance ratio}) better than the previous version: $k$-positive learning ( which is defined as \textbf{batch balance ratio}).
> 3. The intuition under Eq.(13) is we want to assign smaller temperatures for tail classes and larger temperatures for head classes. So for each class $i$, its temperature value $\tau_i$ needs to be proportional to its frequency in the training set. Besides, this $\tau_i$ has to lie in a specific range (to ensure it does not hurt the model performance), therefore we rescale ( by dividing with frequent($class_{max}$)) and shift it into a reasonable range (by $\gamma$).\\
> We propose \textbf{adaptive temperature} as a way to balance the gradient in long-tailed learning. Our main assumption is that the gradient of the head classes is usually larger than the gradient of the tail classes, which can bias the training process. Therefore, we need a mechanism to reweight the gradient so that it is smaller for the head classes and larger for the tail classes. \textbf{Adaptive temperature} achieves this by two means. First, it acts as a reweight parameter in Eq(.10), where a small temperature increases the gradient magnitude and a large temperature decreases it. Second, as we show in Eq(.12), the negative gradient of a sample $x$ in class $i$ is proportional to the number of samples in class $i$. This means that the head classes have a larger negative gradient. However, a large temperature reduces the impact of false hard-negative samples (Eq(.12)), which are the main source of the large negative gradient in the head classes. Thus, \textbf{adaptive temperature} helps balance the negative gradient between the head and tail classes by eliminating the effect of false hard-negative samples.
> 4. Please refer to our comment about the large-scale dataset above.

---

### Review · Reviewer_aM4e · 2023-06-06

**Summary Of Contributions:**

The paper tackles long-tailed recognition using contrastive learning and prototype learning. The authors propose GloCo using k-global positive selection and adaptive temperature. Another method, ProCo, is also introduced to improve prototype learning.

**Audience:**

Yes

**Broader Impact Concerns:**

NA.

**Claims And Evidence:**

Yes

**Requested Changes:**

See above suggestions in weak part.

Additional questions:
1. What is align_i in the Inter_improve formulation?
2. How is GloCo combined with ProCo in training?

**Strengths And Weaknesses:**

Strengths:
1. The paper aims at a specific problem in long-tailed recognition using contrastive learning. The proposed methods are simple and effective.
2. The intuitions of the methods are mathematically proved, which is appreciated. The math derivations also provided insight into the problem.
3. The paper’s ideas are well written and clarified.

Weaknesses:
The main concern is about the experiments.
1. Empirical analysis are insufficient. The paper lacks experiments on larger datasets, such as ImageNet-LT or iNaturalist.
2. While the authors claim their method can outperform the SOTA methods, many other methods, especially contrastive learning ones, yield better performance. Refer to paperswithcode.com for the baselines.
3. Table 1 lacks comparison with other baseline methods. Comparance between the proposed two methods cannot verify anything.
4. Ablation studies on Adaptive temperature and ProCo are missing.
5. Writing could be further improved, such as “balanced feature space” instead of “balance feature space”, “inspired by” instead of “inspiration from” etc.

---

> ### Author Response · Authors · 2023-06-21
> **We thank the reviewer for your positive comment. We would like to address your concerns as below:**
>
> 1. Please refer to our comment about the large-scale dataset above.
> 2. We have updated the results of 2 SOTA methods using contrastive learning PCL [1] and BCL [2] in Tables.3 and Tables.4. Our previous baseline does not include these methods because even though they are contrastive methods, their training strategy is different from ours. First, PCL and BCL are trained on 400 and 800 epochs, respectively. Moreover, they use AutoAugment and Cutout augmentation (which have shown in their experiment that it can boost the performance a lot), while our methods, KCL, and TSC are trained on 200 epochs with MoCo augmentation. Therefore, we believe that our methods can train jointly with BCL and PCL and improve their accuracy.
> \
>  Reference\
> [1]. [zhu2022balanced] Balanced Contrastive Learning for Long-Tailed Visual Recognition,
> Zhu, Jianggang and Wang, Zheng and Chen, Jingjing and Chen, Yi-Ping Phoebe and Jiang, Yu-Gang.\
> [2]. [cui2021parametric] Parametric contrastive learning, Cui, Jiequan and Zhong, Zhisheng and Liu, Shu and Yu, Bei and Jia, Jiaya.
> 3. We have added the baseline result for various metrics. Please refer to Table.1 (which is highlighted in blue
> color) for more detail. In this table, we compare the results of our model with the baseline (KCL). Generally, our model helps the feature space decrease the intra-alignment and increase the inter-uniformity, therefore improving their separable ability.
> 4. We already add an ablation study on both Adaptive temperature and GloCo in section 5.3. The experiment
> results on the setting of Adaptive temperature and ProCo are updated in Table.6, which is highlighted in
> blue color. The table shows that Adaptive temperature, or GloCo when applied individually can help the model learn better.
> 5. We have fixed your proposed error and polished this paper again.\
> Additional questions:
>  1. We have updated this formula in section 4 to be more clear, which is highlighted in green color. The $align_i$ here is actually the $variance$ in the feature space of class $i$.
>  2. Please refer to section 3.3. I have added our join training loss function of GloCo and ProCo, which is highlighted in green color. The final loss is the weighted combination of GloCo and ProCo when we train our model.

---

> ### Comment · Action_Editors · 2023-07-13
> **Official Recommendation**
>
> Dear Reviewer aM4e,
>
> Can you submit your official recommendation so that we can go ahead to the next step? Thanks!
>
> AE

---

### Review · Reviewer_5hQe · 2023-06-07

**Summary Of Contributions:**

This paper proposes two techniques to improve supervised contrastive learning on long-tailed datasets. The first proposed technique called GloCo selects the positive samples globally instead of the minibatch and dynamically schedules the temperature hyperparameter to re-balance the training loss. The second technique called ProCo generates the prototype based the selected coreset and then uses the prototype training to further improve the performance. This paper provides the theoretical analyses of the proposed techniques and the empirical results to support their claim.

**Audience:**

Yes

**Broader Impact Concerns:**

None.

**Claims And Evidence:**

Yes

**Requested Changes:**

Please kindly refer to the comments in the section Weaknesses and make corresponding changes.

**Strengths And Weaknesses:**

Strengths:

1. The authors used a binary classification problem to theoretically analyze the motivation of the proposed method. GloCo can lead to a smaller imbalance factor, thus gaining improved performance.
2. The authors leveraged the existing coreset selection method for generating the effective prototype, which is reasonable.
3. The empirical results seem to show that the proposed method can lead to significant improvement.

Weaknesses:

1. The global selection needs to select the positive samples globally instead of a minibatch, which could consume more running time than baselines. It would be better for the authors to analyze the running time as well.
2. The authors only provide the performance of GloCo and GloCo with ProCo. However, it is still unknown how effective is the proposed ProCo. It would be better for the authors to provide the performance of ProCo.
3. The experimental results are not comprehensive. The authors only used ResNet-32 as the backbone model and CIFAR-10/100 as the training set. I suggest the authors provide the results of various backbone models such as ResNet-50 and WideResNet trained on some large-scale datasets such as ImageNet to support their claim.
4. In Table 1, I suggest the authors provide the results of the baseline. Without the results of baseline, it is hard for me to judge the effectiveness of the proposed method.

---

> ### Author Response · Authors · 2023-06-21
> **We thank the reviewer for your positive comment and your valuable contribution to improving the quality of our paper based on the concerns you have noticed. We would like to address your concerns point by point below.**
>
> 1. We have conducted an ablation study about the running time of different modules in section 5.3 which include KCL (baseline), global selection, adaptive temperature, and ProCo. Please refer the Table.2 for more detail. To sum up, the model takes longer to run when either the number of positive samples in global selections or the size of the coreset in ProCo is increased. Compared to the baseline model, the runtime increased by about 2.3 times with global positive selection(k=3) and by about 1.8 times with ProCo.
> 2. We already add an ablation study on both Adaptive temperature and GloCo in section 5.3. The experiment results on the setting of Adaptive temperature and ProCo are updated in Table.6, which is highlighted in blue color. In general, applying each Adaptive temperature module, or ProCo individually, can improve the performance of the baseline model.
> 3. Please refer to our official comment for the large-scale dataset.
> 4. We have added the baseline result for various metrics. Please refer to Table.1 (which is highlighted in blue color) for more detail. Generally, our model helps the feature space decrease the intra-alignment and increase the inter-uniformity, therefore improving their separable ability.

---

### Comment · Action_Editors · 2023-06-06
**The system said that a reviewer is five days late on a reviewing task**

Dear reviewer aM4e,

Could you please acknowledge that you have noticed this reviewing task? Thank you!

AE

---

> ### Comment · Reviewer_aM4e · 2023-06-06
>
> Thanks, I noticed this review request while recently I have no time to review the frequently asked reviewing work load from TMLR. May I decline this review invite? Thanks!

---

> > ### Comment · Action_Editors · 2023-06-06
> > **Understood**
> >
> > Dear reviewer aM4e,
> >
> > This time, I will replace you with another reviewer for emergency. However, please be aware of the following policy from TMLR.
> >
> > > **Annual Quota.** Reviews are assigned on a rolling basis, as submissions are received. By default, TMLR will not assign you more than six reviews per year. The default quota is 6 papers per year, but it is possible to change it through the openreview console, as well as to mark yourself temporarily unavailable for periods such as illness, vacation or work leave. On the other hand, you are expected to accept all requests for review of submissions that fall within your expertise and quota. Acceptable exceptions include having an unsubmitted review for another TMLR submission.
> >
> > If you are too busy and cannot review for TMLR, you can mark yourself as "unavailable" for a while.
> >
> > AE

---

> > > ### Comment · Reviewer_aM4e · 2023-06-06
> > >
> > > Thanks for letting me know the rule. I will take the reviewing role this time and submit my review within this week.

---

> > > > ### Comment · Action_Editors · 2023-06-06
> > > > **Thanks for your contribution**
> > > >
> > > > Dear reviewer aM4e,
> > > >
> > > > If you don't have enough time, please let me know and I will find another reviewer. Thank you!
> > > >
> > > > AE

---

### Author Response · Authors · 2023-06-21
**Experiment on the large-scale dataset (Imagenet-LT)**

We have conducted experiments on the ImageNet-LT dataset and obtained a minor improvement of 0.3% over the baseline. As we mentioned in the paper (section 5.2, Table.4), the model performance can be improved more after we run cross-validation to select optimal hyperparameters for ImageNet-LT (the current model is trained using the same hyperparameters with CIFAR-100-LT). We are fine-tuning it and will update the results with optimized parameters within a week.
Thank you!

---

### Author Response · Authors · 2023-06-21
**Official Comment by Authors**

Dear reviewers, thank you for your valuable comments. We uploaded a revised manuscript. The changes to the manuscript are indicated in blue and green font. In the comments below we address each reviewer individually.
Thank you!

---

### Comment · Action_Editors · 2023-09-14
**Camera Ready Revision**

Dear authors,

The system says that your camera-ready revision was due on **01 Sept 2023**. Just a friendly reminder.

AE

---

### Decision · Action_Editors · 2023-07-13

**Recommendation:** Accept as is

**Comment:**

The current submission worked on long-tailed visual recognition based on contrastive learning. It has three contributions: global selection of k positive data, adaptive temperature, and coreset selection (which is the prototype learning part). We may regard all of the three contributions as major contributions, but they are relatively independent and the final proposal is the combination of them, making the submission look more like engineering than science. I think the authors did a good job for their rebuttal, as one reviewer said that
> After reading the response to all the reviews, I lean towards accepting the paper for the updated and added experiment results.

On the other hand, two reviewers argued for rejection (one changed his/her mind later after I clarified the acceptance criteria):
> *Official recommendation:* I still think that the proposed theory is insufficient to support the proposed method. Hence, I weakly recommend rejecting this paper.
*Internal discussion:* My major concern is that it is still unclear if Prop. 1 has anything to do with the final performance. The problem comes from the fact that the imbalanced ratio is defined as the ratio of two combinatorial numbers (how many ways to pick out (fixed) k positive instances ) instead of the actual sample frequency. Also the notion of selection space is also quite confusing, why does calculating an imbalance ratio require an extra space?

> *Official recommendation:* Thanks for the authors' responses. Most of my concerns have been solved. However, the improvement on the large-scale datasets is really insignificant and the proposed method is even less efficient since it needs to consumes more running time to do the global positive selection (ref. response 1). Therefore, I lean to rejection.
*Internal discussion:* Thanks for AE sharing the Acceptance Criteria of TMLR. I think the empirical results can support the claims. Therefore, I lean toward acceptance.

According to the evaluation criteria, the novelty/significance is not the main point, but it is whether or not the claims made in the submission are supported by accurate, convincing and clear evidence. I personally think the theory is not a major issue even if it has no business with the final performance (because this is an algorithm paper rather than a theory paper). I also agree with a reviewer that the empirical results can support the claims. Therefore, we should accept the submission for publication.

**Audience:**

Yes.

**Claims And Evidence:**

Yes.

**Resubmission Of Major Revision:**

The authors may consider submitting a major revision at a later time.

---

> ### Author Response · Authors · 2023-09-15
> **Thank you all**
>
> Dear Editors In Chief, Action Editors, and all Reviewers,
>
> We would like to thank you for your valuable and constructive comments. We have updated the camera-ready version of the manuscript.
>
> Sincerely, Authors.